

**Evidence of ketene emissions from petrochemical industries and implications for**
**ozone production potential**
Chinmoy Sarkar[1,a,*], Gracie Wong[1], Anne Mielnik[1], Alex B. Guenther[1], Taehyoung Lee[2], Taehyun
Park[2], Jihee Ban[2], Seokwon Kang[2], Jin-Soo Park[3], Joonyoung Ahn[3], Danbi Kim[3],  Hyunjae Kim[3],
Jinsoo Choi[3], Beom-Keon Seo[4],Jong-Ho Kim[4], Jeong-Ho Kim[5], Soo Bog Park[4], Saewung Kim[1,*]
[1]Department of Earth System Science, University of California Irvine, 92697, California, USA
[2]Department of Environmental Science, Hankuk University of Foreign Studies, Yongin 17035,
South Korea
[3] National Institute of Environmental Research, Inchoen 22689, South Korea
[4]Institute of Environmental Research, Hanseo University, Seosan-si, South Korea
[5]APM Engineering Co. Ltd., Seoul, South Korea
[a]now at: Air Quality Research Center, University of California Davis, One Shields Avenue, Davis,
CA 95616, USA
*Correspondence to: Chinmoy Sarkar (chinmoysarkar8@gmail.com) and Saewung Kim
(saewung.kim@uci.edu)










**Abstract**
Ketene, a rarely measured reactive VOC, was quantified in the emission plumes from Daesan
petrochemical facility in South Korea using airborne PTR-TOF-MS measurements. Ketene mixing
ratios as high as ~ 40 - 50 ppb were observed in the emission plumes. Emission rates of ketene
from the facility were estimated using a horizontal advective flux approach and ranged from 84 –
316 kg h$^{-1}$. These emission rates were compared to the emission rates of major VOCs such as
benzene, toluene, and acetaldehyde. Significant correlations ($r^2 > 0.7$) of ketene with methanol,
acetaldehyde, benzene, and toluene were observed for the peak emissions, indicating commonality
of emission sources. The calculated average ketene OH reactivity for the emission plumes over
Daesan ranged from 3.33 - 7.75 s$^{-1}$, indicating the importance of the quantification of ketene to
address missing OH reactivity in the polluted environment. The calculated average $O_3$ production
potential for ketene ranged from 2.98 – 6.91 ppb h$^{-1}$. Our study suggests that ketene has the
potential to significantly influence local photochemistry and therefore, further studies focusing on
the photooxidation and atmospheric fate of ketene through chamber studies is required to improve
our current understanding of VOC OH reactivity and hence, tropospheric $O_3$ production.
**1. Introduction**
Reactive volatile organic compounds (VOCs) can have atmospheric lifetimes ranging from
minutes to days (Atkinson, 2000) and have significant influence on regional air quality as they
participate in atmospheric chemical reactions that leads to the formation of secondary pollutants
such as tropospheric ozone ($O_3$) and secondary organic aerosol (SOA). Both tropospheric $O_3$ and
SOA are important from the standpoint of air quality and human health and have impact on the
radiative forcing of the atmosphere (IPCC, 2013). In addition, through chemical reactions with the
hydroxyl radicals (major oxidant of the atmosphere; Lelieveld et al. (2004)), photodissociation and
radical recycling reactions, VOCs strongly influence the $HO_x$ (OH, $HO_2$) radical budget that
controls the removal rates of gaseous pollutants from the atmosphere, including most greenhouse
gases (such as $CH_4$).
Ketene (ethenone; CAS: 674-82-8; $H_2C=C=O$) is a highly reactive oxygenated VOC. Recent
studies have revealed that gaseous ketene has very high pulmonary toxicity and can be lethal at
high concentrations (Wu and O'Shea, 2020). Ketene is formed due to pyrolysis reactions of
furfural derivatives and furans emitted during thermal cracking of cellulose and lignin in biomass
material (Kahan et al., 2013). Ozonolysis of propene and oxidation of heterocyclic oxepin
(benzene oxide) also produce ketene in the atmosphere (Klotz et al., 1997;McNelis et al., 1975).
Due to the presence of both double bond and carbonyl functional groups, ketene is highly reactive
and can play a significant role in ambient OH reactivity and hence OH recycling processes
(Lelieveld et al., 2016). Kahan et al. (2013) has demonstrated that hydration reaction of ketene can
form acetic acid under ambient conditions. Therefore, ketene has the potential to explain acetic
acid chemistry in the troposphere, notably near the biomass burning plumes (Akagi et al.,





2012;Yokelson et al., 2009). A recent theoretical study proposed that hydrolysis of ketene
produces acetic acid at a much faster rate in the atmosphere in presence of formic acid (Louie et
al., 2015). This mechanism pathway can facilitate hydrolysis of ketene if it is adsorbed into the
interface of SOA, and therefore can contribute to rapid growth of aerosols even in the absence of
a proper aqueous environment. A quantum chemical study recently showed that ammonolysis
(addition of $NH_3$) of ketene has the potential to produce acetamide ($CH_3CONH_2$) in the
troposphere (Sarkar et al., 2018). Photooxidation of this acetamide can produce isocyanic acid
(HNCO) that has potential health impacts such as cataracts, cardiovascular diseases, and
rheumatoid arthritis as it undergoes protein carbamylation (Wang et al., 2007;Roberts et al.,
2014;Sarkar et al., 2016).
In this study, we present evidence of direct emissions of ketene into the atmosphere from a
petrochemical facility in South Korea, detected by a high sensitivity proton transfer reaction time-
of-flight mass spectrometry (PTR-TOF-MS) technique during an aircraft measurement campaign
conducted in the summer (May-June) and fall (October) of 2019. Emission rate estimation of
ketene using a horizontal advective flux approach adapted from the top-down emission rate
retrieval algorithm (TERRA) (Gordon et al., 2015) with in-situ chemical and meteorological
observations is presented. Finally, an estimation of the OH reactivity and tropospheric $O_3$
production potential of ketene in the emission plumes is provided and their importance is discussed.
**2. Methods**
**2.1 Aircraft campaign**
Airborne VOC measurements were carried out during seven research flights of typically 3-4 hour
duration, conducted in the summer (May-June) and fall (October) of 2019, to characterize
emissions from large industrial facilities (coal power plants, steel mills and petrochemical facilities)
in the Taean Peninsula, located approximately 50 km south of Seoul metropolitan. A PTR-TOF-
MS (model 8000; Ionicon Analytic GmbH, Innsbruck, Austria) was deployed on the Hanseo
University research aircraft (Beechcraft 1900D, HL 5238) for VOC measurements along with a
fast-meteorological sensor (AIMMS 30; Aventech Research Inc.) that is capable of quantifying
aviation such as global positioning information, heading, angle of attack and meteorological data
such as water vapor, temperature, pressure, three dimensional wind field at 10 Hz resolution. To
capture real time emission activity, the research aircraft encircled individual industrial facilities at
a flight altitude of 300-1000 m above ground level. Table 1 provides details of the research flights
with VOC measurements during summer and fall aircraft campaigns while Figure 1 shows
locations of the industrial facilities and research flight tracks.
**2.2 VOC measurements**
VOC measurements were performed over major point and area sources (Daesan petrochemical
facility, Dangjin and Boryoung thermal power plants, Hyundai steel mills and Taean coal power





plants) using a high-sensitivity PTR-TOF-MS that enables high mass resolution with a detection
limit of low ppb to ppt using $H_3O^+$ as reagent ion (Lindinger et al., 1998;Jordan et al., 2009;Sarkar
et al., 2016;Sarkar et al., 2020). The PTR-TOF-MS was operated over the mass range of 21-210
amu at a drift tube pressure of 2.2 mbar and temperature of 60°C (E/N ~ 136 Td) that enabled
collection of VOC data at 1 Hz resolution. Ambient air was sampled continuously through a Teflon
inlet line (OD = 3/8″; length = 3 m) at an inlet flow rate of 100 sccm. To avoid any condensation
effect, inlet line was well insulated and heated to 40°C. Instrumental backgrounds were performed
using ambient air through a VOC scrubber catalyst heated to 350°C (GCU-s 0703, Ionimed
Analytik GmbH, Innsbruck, Austria).
The mixing ratio calculations for methanol, acetaldehyde, benzene and toluene reported in this
study were done by using the sensitivity factors (in ncps ppb$^{-1}$) obtained from the PTR-TOF-MS
calibrations performed using a gravimetric mixture of a 14-component VOC gas standard
((Ionimed Analytik GmbH, Austria at ~ 1 ppm; stated accuracy better than 6% and NIST traceable)
containing methanol, acetonitrile, acetaldehyde, ethanol, acrolein, acetone, isoprene, methyl vinyl
ketone, methyl ethyl ketone, benzene, toluene, o-xylene, chlorobenzene, α-pinene and 1,2-
dichlorobenzene. Calibrations were performed in the range of 2-10 ppb. In order to establish the
instrumental background, VOC-free zero air was generated by passing the ambient air through a
catalytic converter (stainless steel tube filled with platinum-coated glass wool) heated at 350°C.
The measured ion signals were normalized to the primary ion ($H_3O^+$, m/z = 19) as follows (Sarkar
et al., 2016;Sarkar et al., 2020):
$$ncps = \frac{I(RH^+) \times 10^6}{I(H_3O^+)} \times \frac{2}{P_{drift}} \times \frac{T_{drift}}{298.15} \qquad \dots (1)$$

The VOC sensitivities did not show any significant change during the calibrations performed as
the instrumental operating conditions remained constant, which is in agreement to several previous
studies (de Gouw and Warneke, 2007). Table S1 of the supplement lists the sensitivity factors for
methanol, acetaldehyde, benzene and toluene and their estimated limit of detection (LOD),
calculated as the 2σ value while measuring VOC free zero air at 1 Hz resolution (Sarkar et al.,
2016). VOCs for which we do not have any sensitivity factors from the calibrations (e. g. ketene),
concentrations were estimated based on the reaction rate constants as described by de Gouw and
Warneke (2007). A proton transfer reaction rate coefficient of $2.21 \times 10^{-9}$ cm$^3$ s$^{-1}$ was used (Zhao
and Zhang, 2004) to calculate ketene concentrations. The estimated limit-of-detection (LOD) for
ketene was 0.58 ppb. Data acquisition and analysis of the PTR-TOF raw mass spectra was
accomplished using TofDaq (version 1.89; Tofwerk AG, Switzerland) and PTR-MS-viewer
(version 3.2; Ionicon Analytic GmbH, Innsbruck, Austria) softwares, respectively.






## 3. Results and Discussions

### 3.1 Detection of ketene using PTR-TOF-MS

For the identification of VOCs in the raw mass spectra, we followed the protocol described by
Sarkar et al. (2016) and attributed the ion peak detected at m/z 43.018 in the mass scan spectra to
monoisotopic mass of protonated ketene. Absence of any competing shoulder ion peaks between
42.968-43.068 amu (mass width bin of 0.05 amu) indicated no contribution from other ions in this
mass window as shown in Figure S1 of the supplement. The major advantage of using a PTR-
TOF-MS over a conventional PTR-Q-MS (with a quadrupole mass analyzer; Sarkar et al. (2013))
for VOC measurements is the ability of PTR-TOF-MS to separate the isobaric species such as
ketene (measured at m/z 43.018) and propene (measured at m/z 43.054) based on their
monoisotopic masses, allowing us to characterize more VOC species and thus minimize interfering
compounds. With the conventional PTR-Q-MS, both ketene and propene appear at a nominal mass
of m/z 43 and therefore, individual contribution of propene and ketene at m/z 43 remains unknown.
PTR-TOF-MS overcomes this limitation of PTR-Q-MS due to its high sensitivity and a mass
resolution of m/$\Delta$m > 4000, enabling separate detection of ketene (at m/z 43.018) and propene
(m/z 43.054). Detection of propene at m/z 43.054 using a PTR-TOF-MS is well established and
have been reported in several previous studies (Stockwell et al., 2015;Sarkar et al., 2016;Koss et
al., 2018). On the other hand, Ketene has been quantified only recently at m/z 43.018 using PTR-
TOF-MS in the ambient air (Jordan et al., 2009) and in laboratory biomass smoke (Stockwell et
al., 2015). Therefore, propene does not interfere in the detection of ketene using PTR-TOF-MS as
they show separate peaks in the raw mass spectra (Figure S2 of the supplement). Fragmentation
of propanol also results in propene which is detected at m/z 43.054 by PTR-TOF-MS and therefore,
propanol fragmentation does not interfere in the detection of ketene at m/z 43.018. Figure S3 of
the supplement shows the timeseries plot of the corrected ketene measured at m/z = 43.018 (in red)
and propene measured at m/z = 43.054 (in blue) during the research flight conducted on 29 May
morning. It can be seen from the timeseries that we detected propene as well in the emission plumes
from the petrochemical industries. A list of all the VOCs detected in the emission plumes from
petrochemical industries and other industrial facilities during our campaigns will be provided in a
companion paper (in preparation).
Accurate quantification of ketene with PTR-MS technique also depends on the fragmentation of
acetic acid ($CH_3COOH$) and glycolaldehyde ($C_2H_4O_2$) (Karl et al., 2007), parent ion of which is
measured at m/z 61.027 by PTR-TOF-MS (Stockwell et al., 2015;Sarkar et al., 2016). It is not
possible to differentiate structural isomers acetic acid and glycolaldehyde using PTR-TOF-MS,
however, ~ 82% of acetic acid is reported to contribute to the m/z 61.027 signal (Karl et al., 2007).
Fragmentation of this ion can significantly contribute to ketene signal (m/z = 43.018) in the mass
spectra. During our study, the measured ratio between m/z 61.027 and 43.018 outside of the peak
emission cases (Figure 2) was ~ 0.9, which is consistent with the ratio reported in previous studies
at a similar E/N ratio (Hartungen et al., 2004;Haase et al., 2012). This indicates that the





fragmentation of acetic acid and glycolaldehyde results in about half at m/z 61.027 and the
remaining half at m/z 43.018, which is an interference for ketene signal and was subtracted to
obtain the corrected ketene concentrations. Henceforth, we refer to the m/z 43.018 signal, corrected
for the contribution of acetic acid and glycolaldehyde fragments, as ketene in this manuscript.
Figures 2a and 2b show timeseries plots for the mixing ratios of acetic acid and glycolaldehyde
parent ion (m/z = 61.027; in green), ketene fragment (m/z = 43.018; in brown) and the corrected
ketene (in red) during research flights conducted on 29 May morning and 1 June afternoon. It can
be seen from Figure 2a that there were several high peaks of ketene over the Daesan petrochemical
facility on 29 May. Such high peaks of ketene were observed over Daesan during all flights
conducted in the summer and fall (Figures 2b and S4 of the supplement). These high ketene peaks
are entirely absent in the timeseries of m/z 61.027, indicating that acetic acid and glycolaldehyde
fragmentation made a relatively small contribution to the signal detected in the emission plumes
over Daesan. Some peaks were also observed for both m/z 61.027 and m/z 43.018 over Dangjin
coal power plants and Hyundai steel mills during the flights conducted on 29 May and 23 October
(Figures 2a and S4 of the supplement). However, in most cases, these peaks of m/z 43.018
originated from the fragmentation of m/z 61.027, and so they do not contribute to the corrected
ketene signals. This further demonstrates that ketene has fresh emission sources in the plumes over
Daesan petrochemical facility. Since the measured ketene outside the plume was always < 2 ppb
(Figures 2 and S4 of the supplement), contribution from photochemically produced ketene would
be negligible, further indicating direct emission of ketene from the facility.
VOC correlation analyses were performed for the peak values to identify potential emission
sources for ketene at Daesan. Ketene showed strong correlations ($r^2 > 0.7$) with acetaldehyde,
methanol, benzene, and toluene, indicating commonality of emission sources. Figure S5 represents
example correlation plots between these VOCs during 28 October afternoon flight. Many of these
VOCs are emitted during high temperature production processes such as thermal cracking of
ethylene and production of polypropylene in the petrochemical industries (Cetin et al., 2003;Chen
et al., 2014;Mo et al., 2015). Daesan petrochemical facility is a major manufacturer of heat resistant
polypropylene in South Korea and therefore, ketene could potentially be produced during these
high temperature production processes. However, future studies focusing on VOC measurements
in the stacks and analysis using source apportionment models (e.g. USEPA-PMF; Sarkar et al.
(2017)) will improve our understanding of ketene emissions and chemistry at Daesan.
**3.2 Estimation of ketene emission rate using a horizontal advective flux approach**
Emission rates (ERs) of ketene and accompanying VOCs were estimated by integrating the
horizontal advective flux around Daesan petrochemical facility. A two-dimensional cylindrical
screen is created encompassing each facility during each flight. The mixing ratio flux through this
screen is then used to determine the emission rate coming from the interior of the screen. To create
the screen, a single horizontal path surrounding the facility is determined for flight tracks to





represent the horizontal component of the screen. The start of the horizontal path is approximately
set as the south-east corner of the ellipse and progresses in a counter-clockwise direction. The
horizontal path length ($s$) is calculated in meters and as a function of longitude (x) and latitude (y).
Each measurement point within 100 meters of the determined horizontal path is mapped to the
closest point on the horizontal path but retains its altitude ($z$). This creates a set of points on the
cylindrical screen. The measured mixing ratios of each compound, zonal wind ($U$), meridional
wind ($V$) and air density are interpolated to fill areas on the screen to a resolution of $40 \times 20$ meters
($s \times z$). Interpolation is performed using a radial basis function with weights estimated by linear
least squares. The interpolated screens of zonal wind, meridional wind, and air density are used to
calculate the air flux ($E_{air,H}$) through the screen as follows:
$$E_{air,H} = \iint \rho_{air} U_\perp ds dz \qquad \ldots (2)$$

Air density ($\rho_{air}$) is calculated at each flight position from the measured temperature ($T$), pressure
($p$), and percent relative humidity (RH) as described by (Yau and Rogers, 1996):
$$\rho_{air} = \frac{p}{RT(1 + 0.6\chi_{H_2O})}, \chi_{H_2O} = \frac{A_d \varepsilon}{p} \exp\left(\frac{T_d}{B_d}\right) \qquad \ldots (3)$$

where $R = 287.1$ J kg$^{-1}$ K$^{-1}$; $\chi_{H_2O}$ is the water vapor mixing ratio; $A_d = 3.41 \times 10^9$ kPa; $\varepsilon = 0.622$; $B_d$
= 5420 K and $T_d$ is the dew-point temperature calculated using the August-Roche-Magnus
approximation as follows:
$$T_d(T, RH) = \frac{\lambda \left(\ln\left(\frac{RH}{100}\right) + \frac{\beta T}{\lambda + T}\right)}{\beta - \left(\ln\left(\frac{RH}{100}\right) + \frac{\beta T}{\lambda + T}\right)} \qquad \ldots (4)$$

where $\lambda = 243.12$°C and $\beta = 17.62$.
The wind speed normal to the path is calculated as described by Gordon et al. (2015):
$$U_\perp = \frac{V \frac{ds}{dx} - U \frac{ds}{dy}}{\sqrt{\left(\frac{ds}{dx}\right)^2 + \left(\frac{ds}{dy}\right)^2}} \qquad \ldots (5)$$

The mixing ratios of the compounds are interpolated for each point on the screen and combined
with the air flux to calculate the emission rate (ER) of the compounds using the following equation:
$$ER = M_R \iint \chi_C \rho_{air} U_\perp ds dz \qquad \ldots (6)$$





where $\chi_C$ = mixing ratio of VOCs and $M_R$ = ratio between compound molar mass and the molar
mass of air (42.04/28.97 for ketene). The air density ($\rho_{air}$) from the lowest flight track altitude is
approximated with a linear dependence on altitude. $U_\perp$ is the normal wind vector (positive
outwards). Mixing ratios in the areas between the flight track measurements are interpolated using
a radial basis function with weights estimated by linear least square approximation. Interpolated
screens (resolution 40 × 20 m; horizontal × altitude) of U, V wind and air density were then used
to retrieve air flux through the screens. This method is adapted from the TERRA approach
described by Gordon et al. (2015). The mass-balance approach was used to estimate emissions
through the top of the cylinder. Pressure and temperature were assumed to be constant during the
measurement timeframe. To determine emissions through the top of the cylinder, the following
mass-balance approach was used:
$$E_{air,H} + E_{air,V} + E_{air,M} = 0 \quad \dots (7)$$
where, $E_{air,H}$ is the net horizontal air flux, $E_{air,V}$ is the net air flux through the top of the cylinder,
and $E_{air,M}$ is the change in air mass within the volume. We considered constant average pressure
and temperature for the duration of the observations to assume no change in air mass within the
volume. As a result, the air flux through the top of the cylinder can be considered $E_{air,V} = -E_{air,H}$.
The average value of the mixing ratio at the top of the cylinder is multiplied by $E_{air,V}$ to retrieve
emissions through the top of the cylinder.
Figures 3a and 3b depict mixing ratio screens for ketene for the 29 May morning and 28 October
afternoon flights. For both flights, highest ketene mixing ratios ($\chi_{ketene}$) were measured near the
lowest flight path clearly indicating that surface emission sources caused the bulk of the ketene to
be below the flight track. The estimated net ERs (kg h⁻¹) for ketene and accompanying VOCs are
shown in Table 2a. The net ER represents emissions only from the facility and excludes the
contribution of emissions from outside sources that are upwind of the screen. The difference
between the estimated net ER and the ER going out of the screen (Table S2 of the supplement) for
ketene were < 12% for both flights. For 29 May flight, estimated net ketene ER was similar to that
of toluene and ~ 3 times lower than benzene while it was ~ 4 times lower than acetaldehyde. For
28 October flight, estimated net ketene ER was ~ 1.3 times lower and ~ 1.5 times higher than
benzene and toluene, respectively while it was ~ 2 times lower than acetaldehyde. These results
indicate that accurate estimation of ketene emissions from petrochemical facilities could be as
important as some of the major VOCs and therefore, including ketene (a rarely quantified VOC)
to the emission inventory will be a step forward towards effective VOC mitigation strategies.
To address the uncertainty in the extrapolation method below the measurement heights, we have
used two different approaches - ***Approach 1:*** we considered the nature of emissions from the
petrochemical facility being mainly from evaporative sources. As a result, we assumed that the
mixing ratios of ketene increases as it approaches the ground. This is observed when we used the
radial basis function to extrapolate linearly. To quantify the uncertainty in this extrapolation, we



assumed a constant value for heights under the measurement height equal to the mixing ratios at
the lowest observed altitude and defined it as a "constant" case. We assumed that the "constant"
case represents a lower end estimation due to the nature of the evaporative sources from the facility.
Then, we estimated the uncertainty due to ground extrapolation as the percentage change in
emission rates calculated from the linear radial basis function and from the "constant" case. The
estimated uncertainties were < 20% for most cases. For example, for the 29 May morning flight,
the estimated uncertainty was ~ 16% (Figure 4). As expected, this uncertainty is highly dependent
on the vertical position of the plume, with uncertainty being higher in cases where the highest
mixing ratio observed is at the lowest altitude measured. ***Approach 2:*** To assess the accuracy of
the radial basis function interpolation method, plumes resembling the observed plumes were
simulated. The plumes were generated based on a Gaussian distribution of the mixing ratio:
$$\chi(s, z) = \sum_i \exp\left[-\frac{1}{2}\left(\left(\frac{s - s_{o,i}}{\sigma_{s,i}}\right)^2 + \left(\frac{z - z_{o,i}}{\sigma_{z,i}}\right)^2\right)\right] \qquad \dots (8)$$

where, $\chi$ is the mixing ratio, $s_{o,i}$ is the horizontal plume center, $z_{o,i}$ is the vertical plume center,
and $i$ is the plume number. The parameters used for each date are listed in Table S3 of the
supplement. The flight path of each date is used to sample the simulated plume on the screen. The
simulated Gaussian plume is then reconstructed using the radial basis function interpolation based
on the points sampled from the simulated plume. Figure 5 shows the simulated plumes and radial
basis function-interpolated plumes for the 29 May and 28 October flights. The root-mean square
(RMS) and correlation coefficient ($R^2$) values were calculated to compare the simulated plume
with the radial basis function-interpolated plumes. The calculated RMS and $R^2$ values for 29 May
were 0.034 and 0.983, respectively. For 28 October, calculated RMS and $R^2$ values were 0.018
and 0.991, respectively.
**3.3 OH reactivity and O$_3$ production potential**
The OH reactivity of ketene was calculated according to the following equation (Sinha et al., 2012):
$$Ketene\ OH\ reactivity = k_{Ketene+OH}\ [Ketene] \qquad \dots (9)$$

where $k_{Ketene+OH}$ = first-order rate coefficient for the reaction of ketene with OH radicals and
[Ketene] = measured mixing ratio of ketene. The rate coefficient of $3.38 \times 10^{-11}$ cm$^3$ molecule$^{-1}$ s$^{-1}$
$^{-1}$ was used for the reaction of ketene with OH (Brown et al., 1989). For the 29 May and 1 June
flights (summer campaign), calculated average ketene OH reactivity for the emission plumes over
Daesan were 5.42 and 7.75 s$^{-1}$, respectively. The average OH reactivities during research flights
conducted in October (fall campaign) ranged from 3.33 to 7.35 s$^{-1}$. Table 2b shows the calculated
average and maximum OH reactivity and O$_3$ production potential of ketene during seven research
flights. Several previous studies have reported 50% or more missing OH reactivities near industrial
areas (Kim et al., 2011;Ryerson et al., 2003) and showed large uncertainties affecting HO$_x$ budget.



Ambient ketene was not quantified in these studies due to absence of PTR-TOF-MS and the
attribution of nominal mass of m/z 43 (detected by PTR-QMS) only to propene. With PTR-TOF-
MS measurements, it is clear that both propene and ketene can contribute to the nominal mass of
m/z 43. While the rate coefficient of propene with OH radical is about 10% lower ($3 \times 10^{-11}$ and
$3.38 \times 10^{-11}$ cm$^3$ molecule$^{-1}$ s$^{-1}$ for propene (Atkinson et al., 2006) and ketene (Brown et al., 1989),
respectively at 298 K), their chemical reactions with OH would be different since ketene contains
a carbonyl functional group ($H_2C=C=O$) but propene is an alkene ($H_3C-C=CH_2$). Therefore,
quantification of ketene will improve our estimation of the missing OH reactivity.
Tropospheric $O_3$ formation is significantly influenced by VOCs in polluted environments and has
strong impacts on air quality (ability to form photochemical smog), climate (contribution to
radiative forcing), human health (a pulmonary irritant) and can cause decreased crop yields
(Monks et al., 2015;Jerrett et al., 2009). The $O_3$ production potential of ketene was calculated
according to the following equation (Sinha et al., 2012):
$$O_3\ production\ potential = (Ketene\ OH\ reactivity) \times [OH] \qquad ... (10)$$
Average OH radical concentration of $6.2 \times 10^6$ molecules cm$^{-3}$, derived using a reactive plume
model considering NO$_x$ photochemistry (with 255 condensed photochemical reactions) in power-
plant plumes (Kim et al., 2017), was used for the $O_3$ production potential calculation. The
calculated atmospheric lifetime of ketene using this OH concentration was ~ 1.4 h, indicating that
the spatial scale for which ketene would be effective in photochemistry could be at least a few km
(e. g. ~ 10 km assuming horizontal wind speed of 2 m s$^{-1}$). For the 29 May and 1 June flights,
calculated average $O_3$ production potential for ketene in the emission plumes over Daesan were
4.84 and 6.91 ppb h$^{-1}$, respectively. For research flights conducted in October (fall campaign),
average $O_3$ production potential ranged from 2.98 to 6.56 ppb h$^{-1}$ (Table 2b). However, maximum
$O_3$ production potential for ketene at Daesan was 45.70 ppb h$^{-1}$ on 1 June. Due to its fast reaction
rate with OH, ketene can contribute significantly to VOC OH reactivity, and hence $O_3$ production,
and a quantitative understanding of ketene is vital for tropospheric $O_3$ mitigation efforts. Therefore,
it is important to carry out further field and chamber studies to investigate the implications of
ketene photo-oxidation on HO$_x$ chemistry and the atmospheric fate of ketene.
**4. Conclusions**
Ketene, a rare and highly reactive VOC, was identified and quantified using PTR-TOF-MS
technique in the emission plumes of Daesan petrochemical facility in South Korea during aircraft
measurement campaigns conducted in the summer (May-June) and Fall (October) of 2019. Ketene
mixing ratios of ~ 40-50 ppb were measured in the emission plumes. Estimated ketene emission
rates from the facility using a horizontal advective flux approach ranged from 84-316 kg h$^{-1}$.
Ketene emission rates were compared to the estimated emission rates of benzene, toluene, and
acetaldehyde. In most cases, ketene emission rates were comparable to toluene. During peak
emissions, ketene also showed significant correlations ($r^2 > 0.7$) with acetaldehyde, methanol,



benzene, and toluene, indicating emissions of these VOCs occur from common processes. The
petrochemical facility at Daesan is the largest producer of heat resistant polypropylene in South
Korea and the high temperature production processes of polypropylene could be a potential source
of ketene at Daesan. However, future VOC measurement studies focusing on the stack emissions
at Daesan in combination with source apportionment models such as USEPA-PMF will provide
better insights on ketene emissions and chemistry at Daesan petrochemical facility.
For the emission plumes over Daesan, calculated average OH reactivity for ketene ranged from
3.33-7.75 s$^{-1}$. This indicates the importance of the quantification of ketene in the polluted
environment to address the puzzle of missing OH reactivity. During this study, calculated average
ketene O$_3$ production potential ranged from 2.98-6.91 ppb h$^{-1}$. Our study suggests that ketene can
potentially influence local photochemistry. Therefore, future studies focusing on the
photooxidation processes and atmospheric fate of ketene using chamber studies is required to get
a better insight of ketene formation in the atmosphere. Such studies will also improve our current
understanding of VOC-OH reactivity and hence secondary pollutants formation.


### Acknowledgements

The authors thankfully acknowledge this research by the National Strategic Project-Fine Particle
of the National Research Foundation of Korea (NRF) funded by the Ministry of Science and ICT
(MSIT), the Ministry of Environment (ME), and the Ministry of Health and Welfare
(MOHW) (2019M3D8A1067406) and National Institute of Environmental Research (NIER-
RP2019-152) of South Korea for funding and logistical supports. We also appreciate helpful
discussion on the TERRA application generously provided by Andrea Darlington at
Environmental Canada and Mark Gordon at York University, Canada. S. K. would like to
acknowledge a funding support from Brain Pool Program of National Research Foundation Korea
(NRF) Funded by the Ministry of Science ICT (# 2020H1D3A2A01060699)

### Data Availability

The observational data will be available upon the request to the corresponding authors.

### Author Contributions

A. G., T. L., J. A., S. B. P., and S. K. conceptualized the study; C. S., G. W., A. M., T. P., J. B., S.
K., J-S. P., D. K., H. K., J. C., B-K. S., and J-H. K. conducted the field measurements; C. S., G.
W., S. K., and A. G. analyzed the data; A. G., T. L., J. A., S. B. P., and S. K. supervised the research
and administered the project; C. S., G. W., S. K., and A. G. wrote the original draft; All authors


reviewed and edited the manuscript; All authors have given approval to the final version of the
manuscript.

**Conflict of Interest Disclosure**

The authors declare no competing financial interest.

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

**Table 1.** Summary of research flights with VOC measurements during the summer (May-June) and fall (October) aircraft campaigns in South Korea

| Date | Flight No. | Start Time (LT) | End Time (LT) | Facilities Included | Flight Design | Wind Direction |
|---|---|---|---|---|---|---|
| *Summer 2019:* | | | | | | |
| 29 May | 1 | 09:32 | 12:36 | Daesan, Dangjin, Hyundai | Circular spirals at 6 altitudes; 300 - 1100 m | Southwest |
| 1 June | 2 | 13:42 | 16:13 | Boryoung, Daesan, Dangjin, Hyundai | Circular spirals at 6 altitudes; 300 - 1100 m | Wind data not available |
| *Fall 2019:* | | | | | | |
| 23 October | 3 | 13:30 | 16:45 | Boryung, Taean, Daesan, Dangjin, Hyundai | Racetrack and circular spirals at a single altitude ~ 400 m | East |
| 28 October | 4 | 13:38 | 16:48 | Daesan, Hyundai | Racetrack spirals and crosstrack at 2 altitudes; 400 - 600 m | Southwest |
| 29 October | 5 | 08:14 | 11:14 | Daesan, Hyundai | Racetrack and circular spirals at 2 altitudes; 400 – 600 m | West |
| 30 October | 6 | 13:32 | 16:48 | Boryung, Taean, Daesan, Dangjin, Hyundai | Racetrack and circular spirals at | West |





| | | | | | |
|---|---|---|---|---|---|
| | | | | | 3 altitudes; 400 - 1000 m |
| 31 October | 7 | 13:33 | 15:34 | Boryung, Taean, Daesan, Dangjin, Hyundai | Racetrack and circular spirals at 3 altitudes; 400 - 1000 m | Wind data not available |




**Table 2.** a) Net emission rates (kg h$^{-1}$) of ketene, benzene, acetaldehyde and toluene over Daesan petrochemical facility; b) calculated OH reactivity (s$^{-1}$) and O$_3$ production potential (ppb h$^{-1}$) of ketene during emission plumes measured over Daesan petrochemical facility

| a) Research Flights | Ketene (kg h$^{-1}$) | Benzene (kg h$^{-1}$) | Acetaldehyde (kg h$^{-1}$) | Toluene (kg h$^{-1}$) |
|---|---|---|---|---|
| *Summer 2019:* | | | | |
| 29 May Morning | 312 | 917 | 1256 | 314 |
| *Fall 2019:* | | | | |
| 23 October Afternoon | 286 | 146 | -1 | -43 |
| 28 October Afternoon | 316 | 426 | 619 | 210 |
| 29 October Morning | 27 | 241 | 430 | 103 |
| 30 October Afternoon | 84 | 211 | 359 | 102 |

| b) Research Flights | OH reactivity (s$^{-1}$)* | O$_3$ production potential (ppb h$^{-1}$)* |
|---|---|---|
| *Summer 2019:* | | |
| 29 May Morning | 5.42 (33.76) | 4.84 (30.10) |
| 1 June Afternoon | 7.75 (51.24) | 6.91 (45.70) |
| *Fall 2019:* | | |
| 23 October Afternoon | 7.35 (33.33) | 6.56 (29.80) |
| 28 October Afternoon | 5.28 (15.74) | 4.71 (14.00) |
| 29 October Morning | 3.79 (14.77) | 3.38 (13.20) |
| 30 October Afternoon | 3.33 (19.71) | 2.98 (17.60) |
| 31 October Afternoon | 4.56 (8.09) | 4.07 (7.22) |

*Values in the parentheses represents maximum OH reactivity and O$_3$ production potential














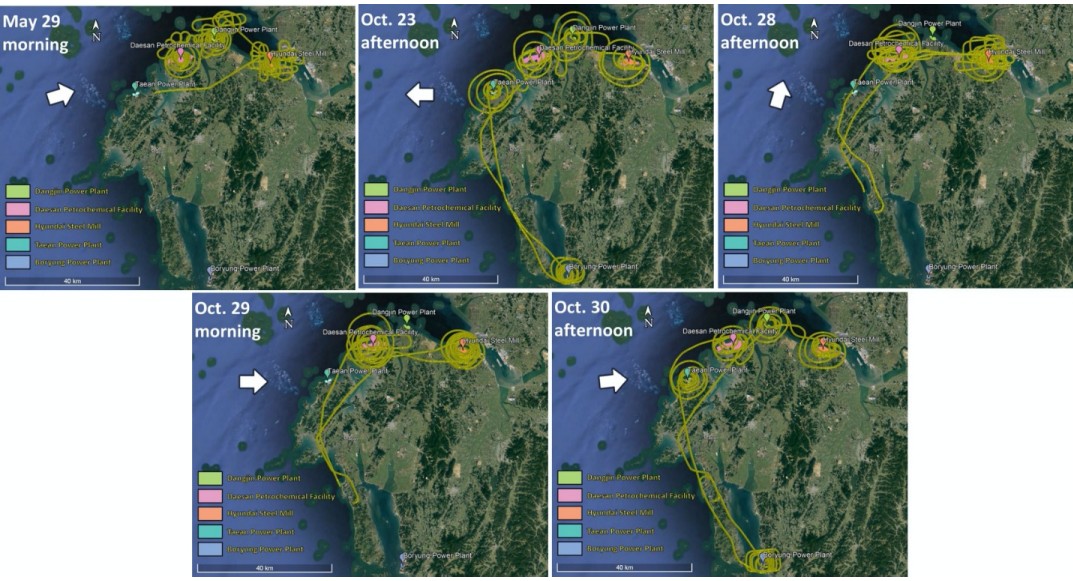


**Figure 1.** Composite ©Google Earth images showing research flight tracks over the Daesan petrochemical facility, Dangjin and Boryoung thermal power plants, Hyundai steel mills and Taean coal power plants during the airborne study conducted in summer (May-June 2019) and fall (October 2019). The white arrow in each plot represents mean wind direction during the flight. Only those flights are shown for which wind direction measurements were available

561

562

563



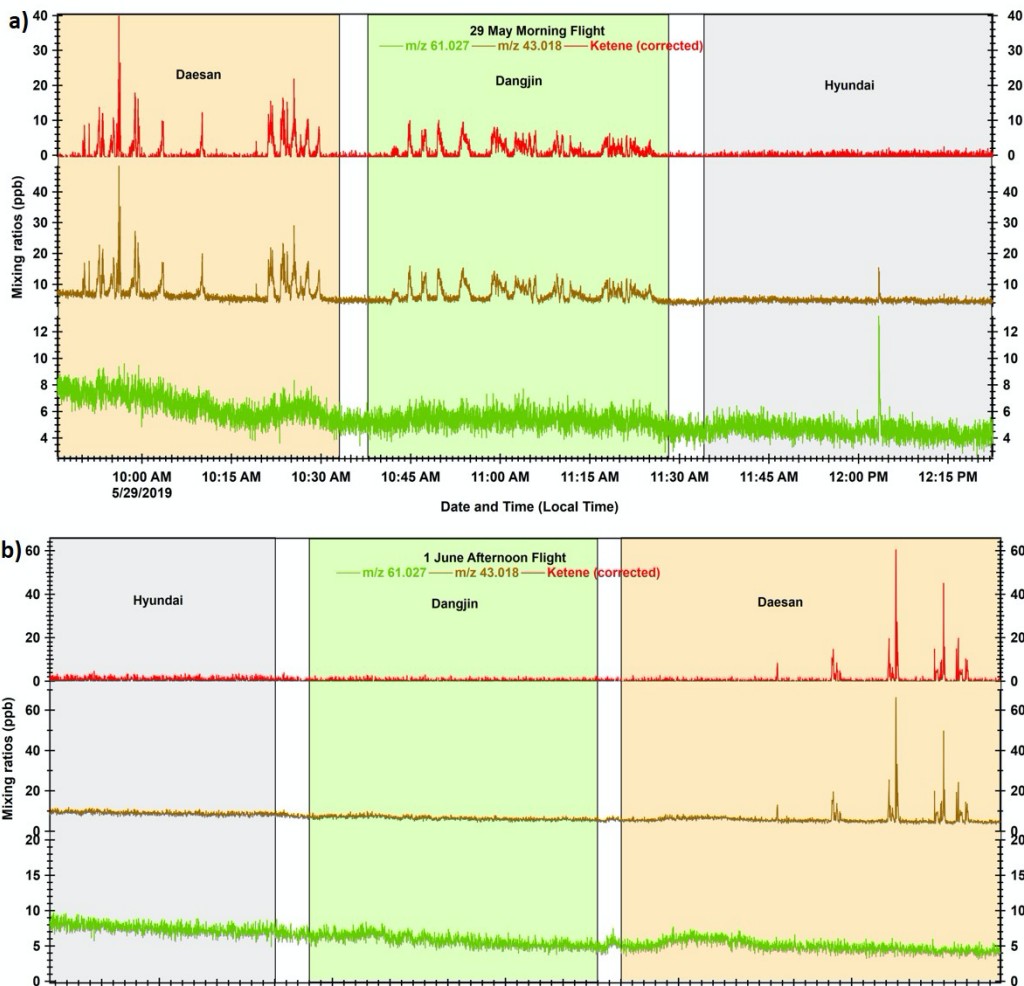

**Figure 2.** Timeseries profiles for mixing ratios (1 Hz resolution) of acetic acid and glycolaldehyde parent ion (m/z = 61.027), ketene fragment (m/z = 43.018) and corrected ketene (corrected for m/z 61.027 fragmentation) during a) 29 May morning flight and b) 1 June afternoon flight. The light pink, light green and light blue shaded areas represent the duration for which the flights were flying over Daesan, Dangjin and Hyundai, respectively





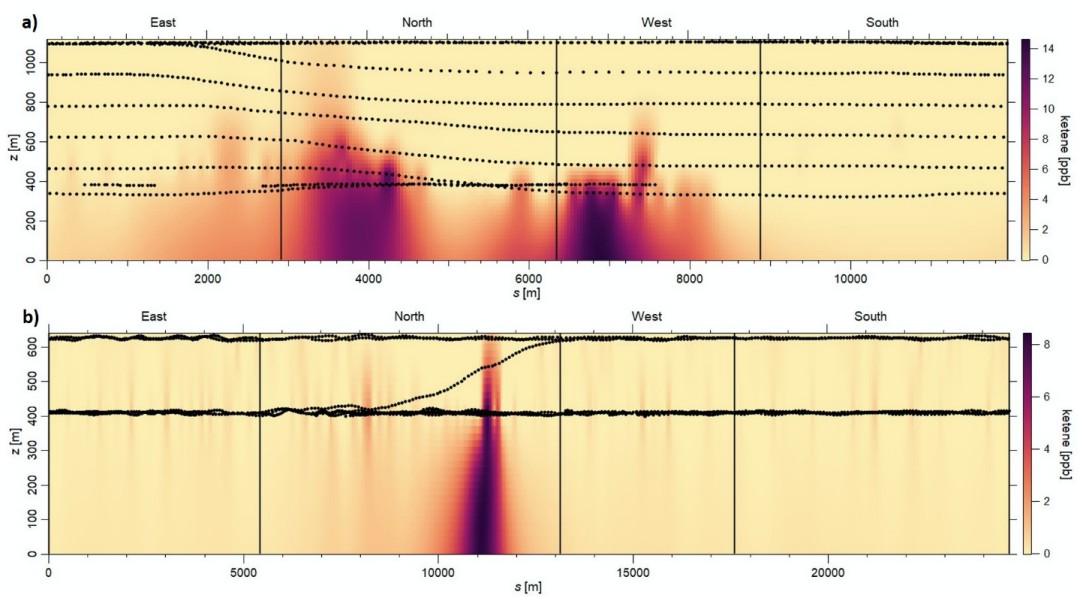


**Figure 3.** Kriging-interpolated ketene mixing ratios for a) 29 May morning and b) 28 October afternoon flights. Black dots represent the flight path.




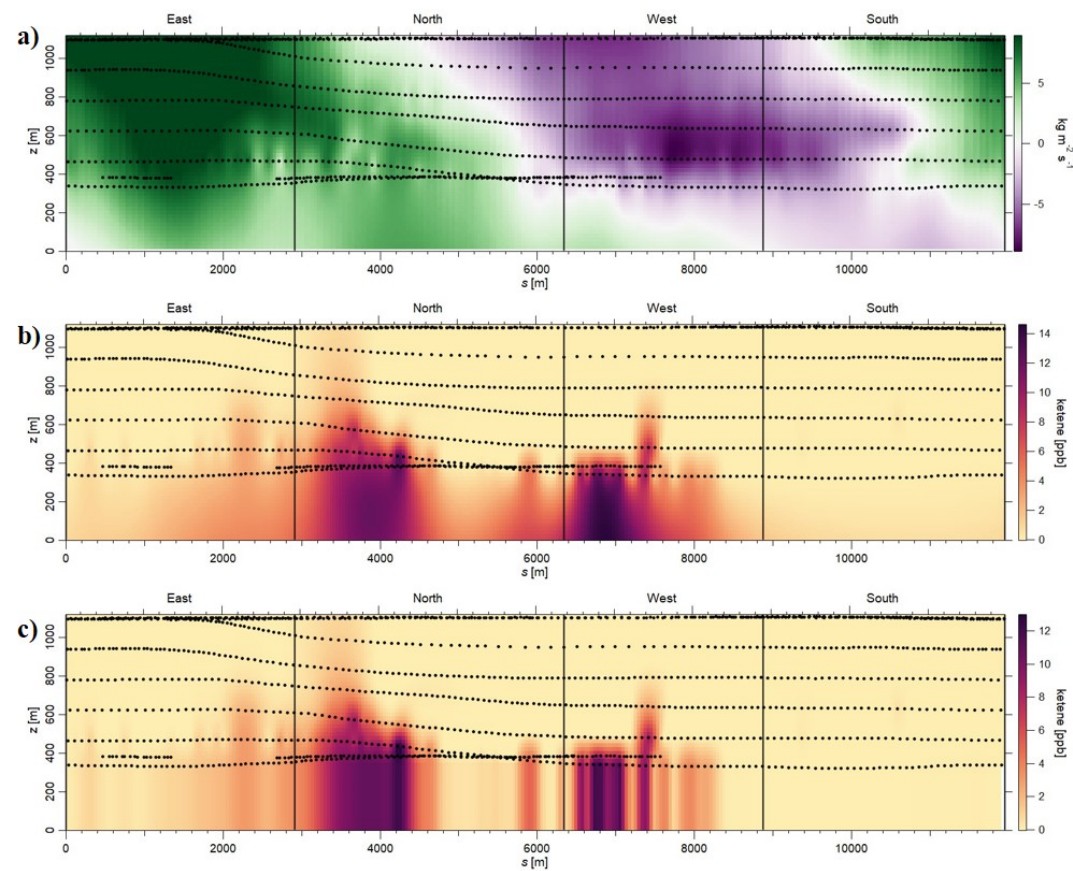


**Figure 4.** Estimation of uncertainty in the emission rates using approach 1 during 29 May morning
flight: a) Air flux screen (green = going out of the facility; purple = into the facility); b) Case 1:
Linear extrapolation using radial basis function (net emission rate from the facility = 312.36 kg h⁻
¹; emission rate going out of the screen = 357.73 kg h⁻¹); c) Case 2: Linear extrapolation using the
"constant case" (net emission rate from the facility = 262.44 kg h⁻¹; emission rate going out of the
screen = 298.74 kg h⁻¹; exponentials are the same for both the cases; uncertainty ~ 15.5% - 16%)






**Figure 5.** a) Simulated plume scenario and b) radial basis function-interpolated plume for May 29
morning flight. c) Simulated plume scenario and d) radial basis function-interpolated plume for
October 28 afternoon flight. Black dots are the flight position measurements of each flight.

