# Peer review of "Evidence of ketene emissions from petrochemical industries and implications for ozone production potential"

_Atmospheric Chemistry and Physics, 2020_

## Short Comment (SC1) · 11 Nov 2020

I am posting a quick comment for saving some time to the reviewers.

I think the assignment of the m/z 43.018 ($CH_3CO^+$) signal to ketene is incorrect. We flew our PTR-TOF-MS instrument on the NASA DC-8 during KORUS-AQ, and we also observed a high m/z 43.018 signal over the Daesan petrochemical complex. The signal was highly correlated with the m/z 87.044 ($C_4H_7O_2^+$) signal and a laboratory study confirmed that the signal ratio was the same as for vinyl acetate. Vinyl acetate produces a strong acetylium ion fragment upon protonation in the PTR-MS analyzers, especially under the PTR-MS operating conditions (high E/N) the authors were using.

Vinyl acetate is expected to be emitted from the ethylene vinyl acetate (EVA) plants at Daesan.

Prof. Armin Wisthaler University of Oslo / University of Innsbruck

---

## Referee Comment (RC1) · Anonymous Referee #1 · 11 Dec 2020

The manuscript presents aircraft measurements of PTR-ToF-MS from a petrochemical facility in South Korea. It applies a horizontal adventive flux approach adapted from the top-down emission rate retrieval algorithm (TERRA) to estimate emission strength. It focuses on a single species, Ketone, and its emissions and impact on OH reactivity and tropospheric ozone formation. The idea about an understudied VOC could be exciting and potentially point to new areas to explore. However, it appears that there are several weaknesses that the paper would need to address.

1. For a study addressing a single species, I'd hope that authors could do a better job in the quantification than just using a proton transfer reaction rate coefficient to estimate

its concentration. The rate coefficient could easily go wrong by more than 200%, and all conclusions would depend on the quality in quantification. Why not do a calibration of ketene in the laboratory? It could also help with the compound identification question raised by Dr. Armin Wisthaler in the Interactive Comment.

2. Specie identification needs to be better addressed, which should be easy to check as a correlation to m/z 87.044, and other potential ions. Again, laboratory experiments could help. Again, I refer to the interactive comment.
* * *

---

## Referee Comment (RC2) · Anonymous Referee #2 · 20 Jan 2021

Sarkar et al. used a PTR-TOF-MS to measure VOCs from the downwind plume of a petrochemical facility in South Korea. They found the signal m/z 43 was high and assigned it to ketene. Using these measurements, they further investigated the environmental implications. The conclusions in this manuscript can only be correct, given the ketene measurement was correct. However, as pointed by Prof. Armin Wisthaler and another reviewer, ketene measurement could be significantly biased. Actually, it might be another species, vinyl acetate, based on measurements in the similar region by Prof. Armin Wisthaler. I strongly agree with Prof. Armin Wisthaler's judgement, as it is well known that m/z 43 or CH3CO+ can be fragmented from a number of compounds, including acetic acid, ethyl acetate. I would recommend the authors consider

the suggestion of Prof. Armin Wisthaler. Even the authors would change the topic from ketene to vinyl acetate, I would also like to see detailed measurements of vinyl acetate by PTR-TOF-MS.

---

## Author Comment (AC2) · 19 May 2021

Evidence of ketene emissions from petrochemical industries and implications for ozone production potential

Chinmoy Sarkar et al. (acp-2020-1103)

We would like to thank Prof. Wisthaler and both the referees for the interactive comments/suggestions on the manuscript. The comments from the referees are in black and our responses are in blue. The texts that have been added to the manuscript are in bold blue. We have also included Sanjeevi Nagalingam and Nicole Jenna Gross as co-authors in the revised manuscript who helped us to carry out laboratory experiments.

**Interactive comment from Prof. Armin Wisthaler:**

I am posting a quick comment for saving some time to the reviewers.

I think the assignment of the m/z 43.018 (CH3CO+) signal to ketene is incorrect. We flew our PTR-TOF-MS instrument on the NASA DC-8 during KORUS-AQ, and we also observed a high m/z 43.018 signal over the Daesan petrochemical complex. The signal was highly correlated with the m/z 87.044 (C4H7O2+) signal and a laboratory study confirmed that the signal ratio was the same as for vinyl acetate. Vinyl acetate produces a strong acetylium ion fragment upon protonation in the PTR-MS analyzers, especially under the PTR-MS operating conditions (high E/N) the authors were using. Vinyl acetate is expected to be emitted from the ethylene vinyl acetate (EVA) plants at Daesan.

Thank you for pointing out the possible contribution of vinyl acetate (m/z 87.044) to the m/z 43.018 signal using PTR-TOF-MS.

We have not observed any high peaks similar to the m/z 43.018 (attributed to ketene) in the timeseries of m87.044 signal. Following figures show the timeseries of m/z 87.044 and m/z 43.018 (ketene) and their correlation plots for the plume episodes (in the inset) during all the flights.

**29 May Morning Flight:**

2

---

## Author Response (AR3)

We would like to thank the Editor (Dr. Chan) for appreciating and highlighting the importance of our work and for recommending the manuscript for publication in ACP subject to minor revisions. We have reworded the manuscript as suggested by the Editor, and these are now reflected in the revised submission. The comments from the Editor are in black and our responses are in blue. The texts that have been modified/added to the manuscript are in bold blue.

Comments to the Author:

Thank you for your interesting manuscript on a potentially important topic in atmospheric chemistry. After considering reviewers' comments, I find that the identification of ketene is not sufficiently strong from an analytical chemistry point of view. The only clear conclusion from the results is that the abundance of m/z 43.018 is very significant. The laboratory investigation to rule out vinyl acetate is useful and convincing. It is, however, difficult to positively identify ketene as the species behind m/z 43.018. (Ruling out VA is useful, but not sufficient.) While ACP is not an analytical chemistry journal, it is still useful to follow some of the principles for chemical identification. The lack of authentic standards or separations, and the possibility of fragmentation in PTR pose serious problems to the identification. In my view, it would take more to convince readers about the ketene identification.

I understand that more experimental work is useful but just drags out the review process. I propose that the authors reword the manuscript and focus on the observations (i. e. abundance of 43.018 ion with a flux comparable to other well-known VOCs) and propose ketene as a possible species, which would be more consistent with the level of certainty in species identification. It is still a useful and important contribution to show that this one ion is representing some unknown species. I am happy to accept a manuscript that conveys this message.

Thank you for the suggestions. We have now reworded several sentences in the manuscript to convey the message that ketene is assigned tentatively as a possible species at m/z 43.018 and future studies are needed to explore the role of this potentially important VOC in the atmosphere. In addition to this, we have changed the title of the manuscript that we believe is best suited for this revised version of the manuscript. The previous title of the manuscript was:

"Evidence of ketene emissions from petrochemical industries and implications for ozone production potential".

The new title is:

**"Unexplored VOC emitted from petrochemical facilities: implications for ozone production and atmospheric chemistry".**

The sentences that were modified in different sections of the manuscript are as follows:

P2 L30-33:

A compound was observed using airborne PTR-TOF-MS measurements in the emission plumes from Daesan petrochemical facility in South Korea. The compound was detected at m/z 43.018 on the PTR-TOF-MS and was tentatively identified as Ketene, a rarely measured

reactive VOC. Estimated Ketene mixing ratios as high as ~ 50 ppb were observed in the emission plumes.

**P2 L42-45:**

Our study suggests that ketene, or any possible VOC species detected at m/z 43.018, has the potential to significantly influence local photochemistry and therefore, further studies focusing on the photooxidation and atmospheric fate of ketene through chamber studies is required to improve our current understanding of VOC OH reactivity and hence, tropospheric O3 production.

**P3 L79-82:**

In this study, we present results from an aircraft measurement campaign conducted in the summer (May-June) and fall (October) of 2019 that shows that a compound emitted from a petrochemical facility in South Korea, was detected at m/z 43.018 by a high sensitivity proton transfer reaction time-of-flight mass spectrometry (PTR-TOF-MS) technique and tentatively identified as ketene.

**P5 L168-170:**

Although PTR-TOF-MS signal at m/z 43.018 could potentially originate from several other VOC species (e. g. acetic acid, glycolaldehyde, vinyl acetate etc.) due to the fragmentation process, our results suggest that ketene is the most probable species detected at this mass during this study.

**P11 L364-368:**

Ketene, a rare and highly reactive VOC, was tentatively identified and quantified as the major species at m/z 43.018 using PTR-TOF-MS technique in the emission plumes of Daesan petrochemical facility in South Korea during aircraft measurement campaigns conducted in the summer (May-June) and Fall (October) of 2019. Ketene mixing ratios of as high as ~ 50 ppb were measured in the emission plumes.

**P12 L387-389:**

Although based on our observation, we strongly believe that the m/z 43.018 signal corresponds to ketene, the possibility of the contribution from vinyl acetate and other species cannot be ruled out completely and therefore, further laboratory and field studies focusing on this aspect are needed.